# Insomnia symptoms and biomarkers of monocyte activation, systemic inflammation, and coagulation in HIV: Veterans Aging Cohort Study

**Brittanny M. Polanka**[1]*, **Suman Kundu**[2], **Kaku A. So-Armah**[3], **Matthew S. Freiberg**[2], **Samir K. Gupta**[4], **Tamika C. B. Zapolski**[5], **Adam T. Hirsh**[5], **Roger J. Bedimo**[6], **Matthew J. Budoff**[7], **Adeel A. Butt**[8,9,10], **Chung-Chou H. Chang**[11], **Stephen S. Gottlieb**[12], **Vincent C. Marconi**[13,14,15], **Julie A. Womack**[16,17], **Jesse C. Stewart**[5]

1 Division of General Internal Medicine, Department of Medicine, Johns Hopkins University School of Medicine, Baltimore, MD, United States of America, 2 Division of Cardiovascular Medicine, Department of Medicine, Vanderbilt University School of Medicine, Nashville, Tennessee, United States of America, 3 Division of General Internal Medicine, Department of Medicine, Boston University School of Medicine, Boston, Massachusetts, United States of America, 4 Division of Infectious Diseases, Department of Medicine, Indiana University School of Medicine, Indianapolis, Indiana, United States of America, 5 Department of Psychology, Indiana University-Purdue University Indianapolis (IUPUI), Indianapolis, Indiana, United States of America, 6 Division of Infectious Diseases, VA North Texas Healthcare System, Dallas, Texas, United States of America, 7 Lundquist Institute, Torrance, California, United States of America, 8 VA Pittsburgh Healthcare System, Pittsburgh, Pennsylvania, United States of America, 9 Weill Cornell Medical College, Doha, Qatar, and New York City, New York, United States of America, 10 Hamad Medical Corp, Doha, Qatar, 11 Department of Medicine, University of Pittsburgh School of Medicine, Pittsburgh, Pennsylvania, United States of America, 12 Department of Medicine, University of Maryland School of Medicine and Baltimore VAMC, Baltimore, Maryland, United States of America, 13 Division of Infectious Diseases, Department of Medicine, Emory University School of Medicine, Atlanta, Georgia, United States of America, 14 Atlanta VA Medical Center, Atlanta, Georgia, United States of America, 15 Department of Global Health, Rollins School of Public Health, Emory University, Atlanta, Georgia, Unites States of America, 16 VA Connecticut Healthcare System, West Haven, Connecticut, United States of America, 17 Yale University School of Nursing, West Haven, Connecticut, United States of America

* bpolank1@jhmi.edu

**Data Availability Statement:** Due to US Department of Veterans Affairs (VA) regulations and our ethics agreements, the analytic data sets

## Abstract

### Background

Insomnia may be a risk factor for cardiovascular disease in HIV (HIV-CVD); however, mechanisms have yet to be elucidated.

### Methods

We examined cross-sectional associations of insomnia symptoms with biological mechanisms of HIV-CVD (immune activation, systemic inflammation, and coagulation) among 1,542 people with HIV from the Veterans Aging Cohort Study (VACS) Biomarker Cohort. Past-month insomnia symptoms were assessed by the item, "Difficulty falling or staying asleep?," with the following response options: "I do not have this symptom" or "I have this symptom and. . ." "it doesn't bother me," "it bothers me a little," "it bothers me," "it bothers me a lot." Circulating levels of the monocyte activation marker soluble CD14 (sCD14),

used for this study are not permitted to leave the VA firewall without a Data Use Agreement. This limitation is consistent with other studies based on VA data. However, VA data are made freely available to researchers with an approved VA study protocol. All data are anonymized prior to being provided to researchers outside of the immediate VA research team. For more information, please visit https://www.virec.research.va.gov or contact the VA Information Resource Center at ViReC@va. gov.

**Funding:** This analysis was funded by grant R01HL126557 from the National Heart, Lung, and Blood Institute/National Institutes of Health awarded to MF, SG, and JS. The Veterans Aging Cohort Study was funded by grant U10AA13566 from the National Institute on Alcohol Abuse and Alcoholism and Veterans Health Administration Public Health Strategic Health Core Group. The funders had no role in study design, data collection and analysis, decision to publish, or preparation of the manuscript.

**Competing interests:** The authors have read the journal's policy and the authors of this manuscript have the following competing interests: RB reports funding from ViiV Healthcare and service on scientific advisory boards for Merck & Co, and ViiV Healthcare. SG reports funding from Gilead Sciences, travel support from Gilead Sciences and Bristol-Myers Squibb, and receipt of advisory board fees from Gilead Sciences and GlaxoSmithKline/ViiV Healthcare. MB reports funding from General Electric. AB reports funding from Gilead Sciences. This does not alter our adherence to PLOS ONE policies on sharing data and materials. There are no patents, products in development or marketed products associated with this research to declare.

inflammatory marker interleukin-6 (IL-6), and coagulation marker D-dimer were determined from blood specimens. Demographic- and fully-adjusted (CVD risk factors, potential confounders, HIV-related factors) regression models were constructed, with log-transformed biomarker variables as the outcomes. We present the exponentiated regression coefficient (exp[b]) and its 95% confidence interval (*CI*).

## Results

We observed no significant associations between insomnia symptoms and sCD14 or IL-6. For D-dimer, veterans in the "Bothers a Lot" group had, on average, 17% higher D-dimer than veterans in the "No Difficulty Falling or Staying Asleep" group in the demographic-adjusted model (exp[b] = 1.17, 95%*CI* = 1.01–1.37, *p* = .04). This association was nonsignificant in the fully-adjusted model (exp[b] = 1.09, 95%*CI* = 0.94–1.26, *p* = .27).

## Conclusion

We observed little evidence of relationships between insomnia symptoms and markers of biological mechanisms of HIV-CVD. Other mechanisms may be responsible for the insomnia-CVD relationship in HIV; however, future studies with comprehensive assessments of insomnia symptoms are warranted.

## Introduction

Although still below that of the general population, the expected lifespan of people living with human immunodeficiency virus (PLWH) has increased considerably with antiretroviral therapy (ART) [1]. However, this increased lifespan has been accompanied by a rise in non-communicable diseases–most notably cardiovascular disease (CVD), a leading cause of death in PLWH [2–4]. In fact, PLWH are at twice the risk of CVD compared to those without HIV [5], and this elevated risk is independent of the risk attributed to HIV, ART, and traditional and non-traditional CVD risk factors [6–8]. Thus, there is a need to identify novel risk factors for CVD in HIV (HIV-CVD) with the potential to serve as future CVD primary prevention targets.

Insomnia is one such possible risk factor for CVD that has been largely ignored in the HIV population. Sleep disturbance is a commonly reported experience among PLWH, with an estimated prevalence of 58% [9]. In addition, PLWH may exhibit an increased risk of developing sleep disturbance compared to the general population [10]. Our recent work using the Veterans Aging Cohort Study (VACS) Survey Cohort was the first to examine insomnia as a predictor of HIV-CVD. Results indicated that PLWH who were bothered a lot by difficulty falling or staying asleep exhibited a 66% greater risk of incident CVD than those without these symptoms, independent of demographics, CVD risk factors, additional potential confounders, and HIV-specific factors [11].

Inclusion of insomnia as a possible risk factor for CVD has received greater attention in the general population, with substantial evidence supporting an insomnia-CVD relationship. Meta-analytic evidence suggests that elevated insomnia symptoms are associated with an increased risk of CVD events (risk ratios [*RR*s] = 1.28–1.55) [12–14]. Furthermore, insomnia has been associated with biological mechanisms that have also been associated with HIV-CVD [15], including altered immune function [16], increased systemic inflammation [17], and

heightened coagulation [18–24] in non-HIV populations. Most striking are findings from the largest randomized control trial of behavioral interventions for insomnia in older adults suggesting a causal link between insomnia and CVD risk. Specifically, Irwin et al. [25] found that older adults receiving cognitive-behavioral therapy for insomnia (CBT-I), versus a sleep education seminar, exhibited a 74% lower risk of elevated C-reactive protein (CRP; $\geq$ 3.0 mg/L), an inflammatory marker implicated in and predictive of the development of CVD [26], at the 16-month follow-up.

Given previous work indicating a link between insomnia and incident CVD in PLWH and the complementary supportive evidence in the general population, it is now important to examine the relationships between insomnia and biological mechanisms that have also been associated with HIV-CVD. Thus, the aim of this study was to determine the associations of insomnia symptoms with biomarkers of immune activation, systemic inflammation, and coagulation among PLWH. We hypothesized that insomnia symptoms would be associated with higher circulating levels of the monocyte activation marker soluble CD14 (sCD14), the inflammatory marker interleukin-6 (IL-6), and the coagulation marker D-dimer.

## Methods

### Study design, setting, and participants

Data for the present study came from the VACS Biomarker Cohort, a cross-sectional subsample of the VACS-9 parent study consisting of participants who provided a blood sample between 2005–2006 [27,28]. VACS-9 is a prospective, multisite, cohort study of HIV-positive veterans and age, sex, race/ethnicity, and clinical site-matched HIV-negative veterans from nine Department of Veterans Affairs (VA) medical centers across the U.S. [29,30]. The institutional review boards of the West Haven VAMC, the VAMC Tennessee Valley Health Care System, and the Vanderbilt University Medical Center IRB approved this study. All participants in the VACS 9 completed a written informed consent. All participants provided written informed consent to have their data from their medical records used for research. From the total VACS Biomarker Cohort sample ($N$ = 2,386), we excluded veterans without HIV ($n$ = 837) and those missing a follow-up date ($n$ = 7). Thus, our final sample consisted of 1,542 HIV-positive veterans (see Table 1 for participant characteristics).

### Measures and procedures

**Exposure variable.** Insomnia symptoms were assessed by the insomnia item of the VACS HIV Symptom Index–a 20-item, self-report questionnaire assessing the frequency and bother of common symptoms in PLWH exposed to multidrug ART and protease inhibitors [31]. The VACS HIV Symptom Index asks participants to indicate what response best describes their experience of each symptom over the past four weeks using the following options: 0 = "I do not have this symptom" or "I have this symptom and. . ." 1 = "it doesn't bother me," 2 = "it bothers me a little," 3 = "it bothers me," or 4 = "it bothers me a lot." We used responses to the insomnia item–"Difficulty falling or staying asleep?"–to create a 5-level insomnia symptoms variable. From this variable, four dummy coded variables were created with the "No Difficulty Falling or Staying Asleep" group as the reference category (i.e., "I do not have this symptom") [11]. We utilized the questionnaire data closest to the blood collection date (median [25th-75th percentile] = 7 [0–133] days prior to blood draw).

**Outcome variables.** Three outcomes were examined: the monocyte activation marker sCD14 (ng/mL), the inflammatory marker IL-6 (pg/mL), and the coagulation marker D-dimer (µg/mL). As is described elsewhere [27,28,32], participant blood samples were collected at one time point between 2005–2006. Serum samples were collected using serum separator and

**Table 1. Characteristics of veterans with HIV in the VACS biomarker cohort ($N$ = 1,542).**

| Demographic Factors | |
|---|---|
| Age, years | 52.2 (8.2) |
| Sex, male | 1,500 (97) |
| Race/Ethnicity | |
| White | 292 (19) |
| African American | 1,065 (69) |
| Hispanic | 127 (8) |
| Other | 58 (4) |
| **CVD Risk Factors** | |
| Pre-existing CVD | 269 (17) |
| Hypertension | |
| None | 438 (28) |
| Controlled | 748 (49) |
| Uncontrolled | 355 (23) |
| Diabetes | 290 (19) |
| BMI, kg/m$^2$ | 25.9 (4.8) |
| Smoking | |
| Never | 370 (24) |
| Current | 772 (50) |
| Former | 398 (26) |
| Statin Use | 463 (30) |
| **Additional Potential Confounders** | |
| Hepatitis C Infection | 724 (47) |
| eGFR, mL/min/1.73m$^2$ | 97.8 (31.6) |
| Hemoglobin, g/dL | 13.9 (1.6) |
| Alcohol Use, hazardous use or abuse/dependence | 653 (42) |
| Cocaine Use, past-year use or abuse/dependence | 324 (21) |
| **Insomnia-Related Factors** | |
| PHQ-9 (no sleep item) | 5.6 (6.6) |
| Non-benzodiazepine Sleep Medication Use | 146 (10) |
| Antidepressant Medication Use | |
| SSRI Use | 665 (43) |
| TCA Use | 393 (26) |
| Miscellaneous Other Use | 652 (42) |
| Efavirenz Medication Use | 364 (24) |
| **HIV-Related Factors** | |
| HIV-1 RNA Level, copies/mL | 20,541.2 (75859.8) |
| CD4$^+$ T Cell Count, mm$^3$ | 443.5 (284.6) |
| ART use | 1,302 (84) |
| **Outcome Variables** | |
| sCD14, ng/mL | 1823.31 (553.12) |
| IL-6, pg/mL | 3.36 (7.48) |
| D-dimer, μg/mL | 0.50 (1.08) |

*Note*. Continuous variables are presented as mean (standard deviation) and categorical variables are presented as *n* (%). Outcome variables are presented in their original, untransformed units.

The following variables include fewer than 1,542 participants because of missing data (*n*, % missing): Hypertension (3, 0.2%), BMI (5, 0.3%), smoking (2, 0.1%), eGFR (1, 0.1%), hemoglobin (1, 0.1%), alcohol use (5, 0.3%), cocaine use (76, 4.9%), PHQ-9 (79, 5.1%), HIV-1 RNA (2, 0.1%), and CD4+ cell count (2, 0.1%).

HIV = human immunodeficiency virus; VACS = Veterans Aging Cohort Study; CVD = cardiovascular disease; BMI = body mass index; eGFR = estimated glomerular filtration rate; PHQ-9 = Patient Health Questionnaire-9; SSRI = selective serotonin reuptake inhibitor; TCA = tricyclic antidepressant; ART = antiretroviral therapy; sCD14 = soluble CD14; IL-6 = interleukin-6.

EDTA tubes and were shipped to a central repository at the Massachusetts Veterans Epidemiology Research and Information Center. Assays were performed at the University of Vermont's Laboratory for Clinical Biochemistry Research. All biomarker measurements used four controls per sample to assess interassay coefficients of variability (CVs). sCD14 was measured with an enzyme-linked immunsorbent assay (Quantikine sCD14 Immunoassay, R&D Systems, Minneapolis, MN) with a detectable range of 40–3,200 ng/mL; CVs ranged from 7.2–8.1%. IL-6 was measured using a chemiluminescent immunoassay (QuantiGlo IL-6 immunoassay, R&D Systems, Minneapolis, MN) with a detectable range of 0.4–10,000 pg/mL; CVs ranged from 7.7–12.3%. D-dimer was measured by a STAR automated coagulation analyzer (Diagnostica Stago) using an immunoturbidometric assay (Liatest D-DI; Diagnostica Stago, Parsippany, NJ) with a detectable range of 0.01–20 ug/mL; CVs ranged from 2.8–14.8%. Because of their positively skewed distributions, all three biomarkers were log-transformed to approximate normal distributions.

**Covariates.** Similar to our previous work, all covariates were determined using self-report measure data or routine clinical care data in the electronic medical record obtained closest to the blood collection date [27]. The following covariates were included, given their prior associations with insomnia, HIV-CVD, or both.

*Demographic factors.* Demographic factors were age, sex (male, female), and race/ethnicity (White, African American, Hispanic, Other).

*CVD risk factors.* CVD risk factors were pre-existing CVD, hypertension, diabetes, body mass index (BMI), smoking, and statin use. Pre-existing CVD was defined by the presence of an *International Classification of Diseases*, *Ninth Revision* (ICD-9) or a *Current Procedural Terminology (CPT)* code prior to the blood collection date for myocardial infarction, unstable angina, coronary heart disease, stroke, congestive heart failure, coronary artery bypass graft, or percutaneous coronary intervention. Hypertension was defined by the average of the three routine outpatient blood pressure values obtained closest to the blood collection date. We created two dummy variables to categorize hypertension based on Joint National Committee on Prevention, Detection, Evaluation, and Treatment of High Blood Pressure thresholds as no hypertension (blood pressure <140/90 mmHg and no antihypertensive medication [reference category]), controlled hypertension (<140/90 mmHg with antihypertensive medication), or uncontrolled hypertension (≥140/90 mmHg) [33]. Diabetes (yes/no) was defined by a validated metric incorporating glucose measurements, diabetes medication use, and/or at least one inpatient or two outpatient ICD-9 codes for diabetes [34,35]. BMI (kg/m$^2$) was defined by one outpatient measurement collected during routine clinical care and was modeled continuously. Smoking (never [reference category], current, or former smoker) was assessed by self-report. Statin use (yes/no) was defined as a filled prescription receipt for a 3-hydroxy-3-methylglutaryl-coenzyme A reductase inhibitor at time of the blood collection.

*Additional potential confounders.* Additional biomedical and behavioral potential confounders were hepatitis C infection, renal function, anemia, alcohol use, and cocaine use. Hepatitis C infection (yes/no) was defined by a positive hepatitis C virus antibody test or at least one inpatient or two outpatient ICD-9 codes for hepatitis C infection [36]. Renal function was defined by estimated glomerular filtration rate (eGFR). Anemia was defined by hemoglobin. Both eGFR and hemoglobin were extracted from the VA Corporate Data Warehouse and modeled continuously. Alcohol use was determined by the Alcohol Use Disorders Identification Test (AUDIT-C) [37] administered closest to the blood collection date and by the presence of alcohol abuse/dependence ICD-9 codes in the electronic medical record prior to the blood collection date. We combined these two data sources into one dichotomous variable as no current use or non-hazardous use versus hazardous use (AUDIT-C ≥ 4) or alcohol abuse/dependence disorder. Cocaine use was assessed by self-report and by the presence of cocaine

use disorder ICD-9 codes in the electronic medical record prior to the blood collection date. We combined these two data sources into one dichotomous variable as never tried or no use in past year versus use in the past year or cocaine abuse/dependence disorder.

*HIV-related factors.* The HIV-related factors were HIV-1 RNA level, CD4$^+$ T cell count, and ART use. HIV-1 RNA level and CD4$^+$ T-cell count, measured during routine outpatient visits, were determined from the VA Corporate Data Warehouse data obtained closest to the blood collection date and modeled continuously. ART use (yes/no) was defined as a filled prescription receipt for any ART closest to the blood collection date (-180 days to +7 days of date).

*Insomnia-related factors.* The insomnia-related factors were depressive symptoms, non-benzodiazepine sleep medication use, antidepressant medication use, and efavirenz medication use. Given the overlap between insomnia and depression, depression symptoms were included in order to examine insomnia's independent associations with the biomarkers. Antidepressant medication use was included, as these medications are commonly prescribed for sleep disturbance [38,39] and have been associated with systemic inflammation among PLWH [40]. Efavirenz medication use was included as these medications are commonly associated with insomnia [41]. Depressive symptoms were measured by the Patient Health Questionnaire-9 (PHQ-9) [42]. We removed item 3 ("Trouble falling or staying asleep, or sleeping too much") from the PHQ-9 total score calculation and subsequently refer to it as PHQ-9 (no sleep item). Non-benzodiazepine sleep medication use (yes/no) was defined as a filled prescription receipt closest to the blood collection date (-ever to +180 days of date) for the following medications: zolpidem, zaleplon, eszopiclone, and indiplon. Antidepressant medication use was defined as a filled prescription receipt for an antidepressant medication closest to the blood collection date (-ever to +180 days of date). We computed three dichotomous variables (yes/no) based on the antidepressant medication type–serotonin reuptake inhibitor (SSRI), tricyclic antidepressant (TCA), and miscellaneous other antidepressant. Of note, the treatment indications for non-benzodiazepine and antidepressant medications are not known. Efavirenz medication use (yes/no) was determined in a similar fashion to that of the ART use variable.

## Statistical analysis

Descriptive statistics–i.e., mean (standard deviation) and frequency count (%)–for the participant characteristics and the biomarker levels across insomnia symptom categories were computed.

Multivariate linear regression models were constructed to estimate the associations between insomnia symptoms and monocyte activation, inflammatory, and coagulation markers in HIV-positive veterans. Two models were constructed for each outcome variable (log-transformed sCD14, IL-6, and D-dimer). Model 1 (demographics-adjusted) consisted of the four insomnia symptom dummy coded variables, age, sex, and race/ethnicity. Model 2 (fully-adjusted), which was our primary model, included the Model 1 variables plus the CVD risk factors (pre-existing CVD, hypertension, diabetes, BMI, smoking, and statin use), the additional potential confounders (hepatitis C infection, renal function [eGFR], anemia [hemoglobin], alcohol use, and cocaine use), and the HIV-related factors (HIV-1 RNA level, CD4$^+$ T cell count, and ART use). Due to the use of log-transformed outcome variables, we present the exponentiated regression coefficient [exp(b)] and its 95% confidence interval (*CI*) for each association of interest. We interpret the percent change in each biomarker per 1-unit increase in the insomnia symptoms dummy coded variables (i.e., switching from the reference category "No Difficulty Falling or Staying Asleep" to the respective insomnia symptom category) using the following equation: [exp(b)-1] x 100.

Four sensitivity analyses were conducted to individually examine the potential influence of the insomnia-related variables on the associations of interest. These three models were constructed for each outcome variable as follows: Model 3: Model 2 plus PHQ-9 total score (no sleep item), Model 4: Model 2 plus non-benzodiazepine sleep medication use, Model 5: Model 2 plus antidepressant medication use (SSRI, TCA, and miscellaneous other use), and Model 6: Model 2 without ART use variable plus efavirenz use.

All analyses were performed using R software (version 3.5.2; www.r-project.org). To address missingness, data examined in regression models underwent multiple imputations using chained equations (MICE) with five separate imputed datasets generated based on predictive mean matching using the 'mice' library of R programming language. Regression models were fit in each imputed dataset and finally combined to obtain pooled effect sizes and standard errors using Rubin's rule [43].

## Results

### Participant characteristics

The characteristics of participants are presented in Table 1. The mean age of our sample of 1,542 HIV-positive veterans was 52 years, with the majority being male (97%) and African American (69%). Participants exhibited high CVD risk factor burden, particularly hypertension (72%) and current smoking (50%). Also of note was our sample's high prevalence of hepatitis C co-infection at 47% and substance misuse/abuse (42% and 21% for alcohol and cocaine, respectively). Regarding insomnia-related variables, participants exhibited low use of non-benzodiazepine medications (10%) but high use of antidepressant medications (26–43%). While the mean PHQ-9 (no sleep item) score of the sample was 5.6, 21% of our sample exhibited clinically-elevated depressive symptoms (i.e., PHQ-9 total score $\geq$ 10) [44]. The distribution of participants across the insomnia symptom categories is presented in Table 2, with each category's respective log-transformed unit mean (standard deviation) for sCD14, IL-6, and D-dimer. Visual inspection of Table 2 suggests a large proportion of participants denied having insomnia symptoms (43%) while approximately 46% of our sample endorsed some level of bother with insomnia symptoms.

### Insomnia symptoms and sCD14

As is shown in Table 3, the dummy coded variables comparing "Doesn't Bother," "Bothers a Little," "Bothers," and "Bothers a Lot" categories to the "No Difficulty Falling or Staying

**Table 2. Biomarker levels across insomnia symptom categories of veterans with HIV in the VACS biomarker cohort.**

| Insomnia Symptoms | sCD14 | IL-6 | D-dimer |
|---|---|---|---|
| No Difficulty Falling or Staying Asleep (*n* = 652) | 7.45 (0.30) | 0.80 (0.82) | -1.30 (0.95) |
| Doesn't Bother (*n* = 176) | 7.47 (0.29) | 0.81 (0.70) | -1.26 (1.20) |
| Bothers a Little (*n* = 291) | 7.48 (0.31) | 0.81 (0.77) | -1.26 (0.99) |
| Bothers (*n* = 182) | 7.48 (0.29) | 0.84 (0.71) | -1.18 (0.99) |
| Bothers a Lot (*n* = 220) | 7.49 (0.24) | 0.86 (0.70) | -1.20 (0.97) |

*Note*. Outcome variables are presented as mean (standard deviation) in their log-transformed units. A total of 21 cases are not included in Table 2, as they were imputed due to missing data on the insomnia symptoms item of the VACS HIV Symptom Index.

HIV = human immunodeficiency virus; VACS = Veterans Aging Cohort Study; sCD14 = soluble CD14; IL-6 = interleukin-6.

**Table 3. Associations of insomnia symptoms categories with sCD14, IL-6, and D-dimer levels in veterans with HIV in the VACS biomarker cohort (N = 1,542).**

| Model | sCD14 | | | IL-6 | | | D-dimer | | |
|---|---|---|---|---|---|---|---|---|---|
| | exp(b) | 95%CI | p-value | exp(b) | 95%CI | p-value | exp(b) | 95%CI | p-value |
| **Model 1: Demographic-adjusted** | | | | | | | | | |
| No Difficulty Falling or Staying Asleep | Ref. | — | — | Ref. | — | — | Ref. | — | — |
| Doesn't Bother | 1.01 | (0.96–1.06) | 0.68 | 0.99 | (0.88–1.13) | 0.93 | 1.06 | (0.89–1.25) | 0.52 |
| Bothers a Little | 1.02 | (0.98–1.07) | 0.27 | 1.02 | (0.92–1.13) | 0.71 | 1.04 | (0.90–1.21) | 0.56 |
| Bothers | 1.03 | (0.98–1.08) | 0.23 | 1.07 | (0.94–1.22) | 0.30 | 1.18 | (0.99–1.39) | 0.06 |
| Bothers a Lot | 1.03 | (0.99–1.08) | 0.14 | 1.10 | (0.98–1.24) | 0.11 | **1.17*** | **(1.01–1.37)** | **0.04** |
| **Model 2: Fully-adjusted** | | | | | | | | | |
| No Difficulty Falling or Staying Asleep | Ref. | — | — | Ref. | — | — | Ref. | — | — |
| Doesn't Bother | 1.00 | (0.96–1.05) | 0.91 | 0.96 | (0.85–1.08) | 0.46 | 1.05 | (0.90–1.23) | 0.54 |
| Bothers a Little | 1.01 | (0.97–1.05) | 0.58 | 0.97 | (0.87–1.07) | 0.51 | 1.03 | (0.91–1.18) | 0.62 |
| Bothers | 1.01 | (0.96–1.05) | 0.72 | 0.99 | (0.88–1.11) | 0.83 | 1.11 | (0.95–1.30) | 0.20 |
| Bothers a Lot | 1.01 | (0.97–1.05) | 0.57 | 1.00 | (0.89–1.11) | 0.96 | 1.09 | (0.94–1.26) | 0.27 |
| **Model 3: PHQ-9 (no sleep item)** | | | | | | | | | |
| No Difficulty Falling or Staying Asleep | Ref. | — | — | Ref. | — | — | Ref. | — | — |
| Doesn't Bother | 1.00 | (0.96–1.05) | 0.99 | 0.96 | (0.85–1.08) | 0.53 | 1.04 | (0.89–1.23) | 0.60 |
| Bothers a Little | 1.00 | (0.97–1.05) | 0.74 | 0.98 | (0.88–1.08) | 0.67 | 1.02 | (0.89–1.17) | 0.76 |
| Bothers | 1.01 | (0.96–1.05) | 0.93 | 1.01 | (0.89–1.14) | 0.93 | 1.09 | (0.92–1.29) | 0.33 |
| Bothers a Lot | 1.00 | (0.95–1.05) | 0.93 | 1.03 | (0.90–1.17) | 0.70 | 1.05 | (0.89–1.25) | 0.56 |
| **Model 4: Non-Benzodiazepine Sleep Medication Use** | | | | | | | | | |
| No Difficulty Falling or Staying Asleep | Ref. | — | — | Ref. | — | — | Ref. | — | — |
| Doesn't Bother | 1.00 | (0.96–1.05) | 0.91 | 0.96 | (0.85–1.08) | 0.46 | 1.05 | (0.90–1.23) | 0.54 |
| Bothers a Little | 1.01 | (0.97–1.05) | 0.58 | 0.97 | (0.88–1.07) | 0.52 | 1.04 | (0.91–1.18) | 0.61 |
| Bothers | 1.01 | (0.96–1.05) | 0.72 | 0.99 | (0.88–1.11) | 0.84 | 1.11 | (0.95–1.30) | 0.20 |
| Bothers a Lot | 1.01 | (0.97–1.05) | 0.58 | 1.00 | (0.89–1.12) | 0.99 | 1.09 | (0.94–1.26) | 0.27 |
| **Model 5: Antidepressant Medication Use** | | | | | | | | | |
| No Difficulty Falling or Staying Asleep | Ref. | — | — | Ref. | — | — | Ref. | — | — |
| Doesn't Bother | 1.01 | (0.96–1.05) | 0.82 | 0.97 | (0.86–1.09) | 0.57 | 1.03 | (0.88–1.21) | 0.73 |
| Bothers a Little | 1.01 | (0.97–1.05) | 0.59 | 0.97 | (0.87–1.07) | 0.52 | 1.01 | (0.88–1.15) | 0.90 |
| Bothers | 1.01 | (0.97–1.06) | 0.67 | 0.99 | (0.88–1.12) | 0.94 | 1.06 | (0.90–1.25) | 0.48 |
| Bothers a Lot | 1.01 | (0.97–1.06) | 0.56 | 1.00 | (0.89–1.12) | 0.97 | 1.04 | (0.89–1.21) | 0.66 |
| **Model 6: Efavirenz Medication Use** | | | | | | | | | |
| No Difficulty Falling or Staying Asleep | Ref. | — | — | Ref. | — | — | Ref. | — | — |
| Doesn't Bother | 1.00 | (0.96–1.04) | 0.99 | 0.96 | (0.85–1.08) | 0.45 | 1.04 | (0.89–1.23) | 0.59 |
| Bothers a Little | 1.01 | (0.97–1.05) | 0.67 | 0.97 | (0.87–1.07) | 0.50 | 1.02 | (0.90–1.17) | 0.72 |
| Bothers | 1.01 | (0.96–1.05) | 0.74 | 0.99 | (0.88–1.11) | 0.82 | 1.11 | (0.94–1.30) | 0.21 |
| Bothers a Lot | 1.01 | (0.97–1.06) | 0.53 | 1.00 | (0.89–1.11) | 0.96 | 1.08 | (0.93–1.25) | 0.29 |

*Note.* For the results presented in Table 3, the outcome variables are log-transformed.

Model 1: Demographic-adjusted (age, sex, race/ethnicity).

Model 2: Fully-adjusted (Model 1 + CVD Risk Factors: Pre-existing CVD, hypertension, diabetes, BMI, smoking, statin use; Additional Potential Confounders: Hepatitis C infection, renal function, anemia, alcohol use, cocaine use; HIV-Related Factors: HIV-1 RNA viral load, CD4+ T cell count, ART use).

Model 3: PHQ-9 (Model 2 + PHQ-9 [no sleep item]).

Model 4: Non-Benzodiazepine Sleep Medication Use (Model 2 + non-benzodiazepine sleep medication use).

Model 5: Antidepressant Medication Use (Model 2 + SSRI use, TCA use, and miscellaneous other antidepressant use).

Model 6: Efavirenz Medication Use (Model 2 [without ART use] + efavirenz use).

*p < 0.05.

sCD14 = soluble CD14; IL-6 = interleukin-6; HIV = human immunodeficiency virus; VACS = Veterans Aging Cohort Study; PHQ-9 = Patient Health Questionnaire-9; CVD = cardiovascular disease; BMI = body mass index; ART = antiretroviral therapy; SSRI = serotonin selective reuptake inhibitor; TCA = tricyclic antidepressant; Ref. = reference category.

Asleep" reference category were not significant in Model 1 (demographic-adjusted; *p*-value range: 0.14–0.68) or Model 2 (fully-adjusted; *p*-value range: 0.57–0.91). Results remained consistent in the sensitivity analyses individually adjusting for PHQ-9 (no sleep item; Model 3), non-benzodiazepine sleep medication use (Model 4), antidepressant medication use (Model 5), or efavirenz medication use (Model 6).

### Insomnia symptoms and IL-6

Similar to the sCD14 results, all of the insomnia symptom dummy coded variable comparisons were not significant in Model 1 (*p*-value range: 0.11–0.93) or Model 2 (*p*-value range: 0.46–0.96; see Table 3). Results remained consistent in the sensitivity analyses individually adjusting for the insomnia-related factors (Models 3–6).

### Insomnia symptoms and D-dimer

Again, the dummy coded variables comparing "Doesn't Bother," "Bothers a Little," and "Bothers" to the "No Difficulty Falling or Staying Asleep" reference category were not significant in Models 1–6 (*p*-value range: 0.06–0.90). However, the dummy coded variable comparing the "Bothers a Lot" category to the "No Difficulty Falling or Staying Asleep" reference category was significant in Model 1 (exp[b] = 1.17, 95% *CI*: 1.01–1.37, *p* = 0.04) adjusting for demographic factors. The exp(b) of 1.15 indicates that a 1-unit change in insomnia symptoms category (i.e., moving from "No Difficulty Falling or Staying Asleep" to "Bothers a Lot") was associated with a 17% increase in D-dimer on average [(1.17–1) x 100] while controlling for demographic factors. However, this association was attenuated and not significant in Model 2 (exp[b] = 1.09, 95%*CI*: 0.94–1.26, *p* = 0.27) adjusting for additional potential confounders. Results for this comparison remained non-significant in the sensitivity analyses individually adjusting for the insomnia-related factors (Models 3–6).

## Discussion

Our examination of the VACS Biomarker Cohort data did not support our hypotheses that greater insomnia symptoms would be associated with higher circulating levels of markers of monocyte activation, systemic inflammation, and coagulation in PLWH. All but one of the tested associations between insomnia symptom categories and sCD14, IL-6, and D-dimer were not significant. Furthermore, the one significant association observed in a demographic-adjusted model–i.e., higher D-dimer among veterans bothered a lot by difficulty falling or staying asleep–was no longer significant in our primary model further adjusting for CVD risk factors, additional potential confounders, and HIV-related factors. Taken together, the present results suggest that insomnia symptoms may not be associated with the biological mechanisms also associated with HIV-CVD examined in the present study.

To our knowledge, this is the first study to investigate associations between insomnia symptoms and immune activation or coagulation markers among PLWH. Others have examined the relationship between sleep parameters or sleep disturbance and systemic inflammation, reporting mixed results. Of the four available studies, two examined group differences, one using self-reported sleep onset latency (SOL) ≤ 30 or > 30 minutes [45] and one using objectively measured SOL, wake-after-sleep-onset (WASO), time of sleep onset, total sleep time (TST), and sleep efficiency (SE) split at the sample median for each variable [46]. Tests of group differences between these two studies yielded conflicting results. Gay et al. [45] found no group differences in various inflammatory markers (i.e., CRP, IL-1beta, IL-2, IL-6, IL-10, IL-13, and tumor necrosis factor [TNF]-alpha) between those with SOL ≤ 30 and > 30 minutes. Wirth et al. [46] found significantly higher or trending higher CRP and IL-6 among those

with indicators of poorer sleep quantity and quality (i.e., lower TST, later sleep onset time, lower SE, higher SOL, and higher WASO). In addition, three of the four studies examined linear or correlational associations. Gay et al. [45] observed significant or trending positive associations between IL-13 and SOL > 30 minutes (a 1-*SD* increase in IL-13 was associated with a 34% increased odds of SOL > 30 minutes, *p* = .024) and IL-10 (a 1-*SD* increase in IL-10 was associated with a 27% increased odds of SOL > 30 minutes, *p* = .068), controlling for genomic estimates of ancestry, race/ethnicity, and viral load. SOL > 30 minutes was not associated with IL-1beta, IL-2, IL-6, TNF-alpha, and CRP. Lee et al. [47] observed positive correlations between objectively measured WASO and CRP (rho = .135, *p* = .023) and TNF-alpha (rho = .121, *p* = .042), a trending negative association between WASO and IL-13 (rho = -.111, *p* = .061), and null associations between WASO and IL-1beta, IL-2, IL-6, and IL-10. Moore et al. [48] observed negative correlations between self-reported sleep disturbance and TNF-gamma (rho = -.697, *p* = .017) and TNF-alpha (rho = -.697, *p* = .017) but not for IL-6 among women, with no significant associations observed among men. Differences in results between the four available studies and our own may be due to differences in the definition of the sleep disturbance variables (e.g., subjective versus objective measurement) and/or sample characteristics (e.g., percentage with undetectable versus detectable viral loads). Of note, because Gay et al. [45] and Lee et al. [47] utilized similar samples from The Symptoms and Genetic Study, our current knowledge of the insomnia-inflammation relationship, including the present study, is based on only four samples of PLWH.

The lack of associations between insomnia and putative biological mechanisms of HIV-CVD in the present study raises two possibilities. The first possibility is our null results reflect the state of nature. If accurate, this would suggest that other mechanisms underlie the previously observed insomnia-CVD relationship among PLWH. Other potential mechanisms include hypertension and overweight/obesity, which consist of both biological and behavioral components. Regarding hypertension, meta-analytic evidence in the general population suggests that adults with individual symptoms of insomnia have a 14–21% increased risk of hypertension [49]. To our knowledge, there are no published studies examining longitudinal associations between sleep variables and hypertension or blood pressure among PLWH. Elucidating insomnia's role in hypertension in the HIV population could be of great importance, given hypertension's high prevalence (19–52%) and its ties to pathological consequences of the HIV virus (e.g., chronic inflammation) and ART treatment (e.g., mitochondrial toxicity and insulin resistance) [50]. Concerning overweight/obesity, general population data suggests that sleep restriction is positively related to subjective hunger, caloric intake, and weight gain and negatively related to insulin sensitivity [51]. A few existing studies have observed a positive association between sleep disturbance and weight variables among PLWH (i.e., obesity and increased waist size) [52,53], although it has not been a focus of the HIV literature thus far. Other potential behavioral mechanisms identified in the general population that may translate to the HIV population include decreased physical activity [54], smoking [55], and poor diet [56].

The second possibility is our null results are due to methodological factors and may or may not reflect the state of nature. For instance, our assessment of insomnia relied on a single item that assessed only difficulty falling or staying asleep within the past four weeks, missing symptoms such as early morning awakening, insomnia-related impairment and/or distress, and insomnia symptom frequency and duration. This reduced content validity could contribute to the misclassification of cases (e.g., participants experiencing symptoms of insomnia are inappropriately classified as not experiencing insomnia). Future research endeavors on insomnia in the HIV population should consider incorporating comprehensive and validated assessments of insomnia symptoms and/or insomnia disorder. To assess insomnia symptoms, we

recommend the Insomnia Severity Index (ISI), which is sensitive to treatment response and can be administered online [57,58]. To assess insomnia disorder, we recommend either the Structured Clinical Interview for Diagnostic and Statistical Manual of Mental Disorders, Fifth Edition (DSM-5) Sleep Disorders (SCISD) interview [59] or the Diagnostic Interview for Sleep Patterns and Disorders (DISP) based on The International Classification of Sleep Disorders, second edition (ICSD-2) criteria [60].

In addition to our study's strengths (large sample, extensive adjustment for potential confounders, and inclusion of monocyte activation and coagulation biomarkers), there are important limitations worth considering. First, the cross-sectional, observational study design prevents examination of the directionality of associations between insomnia symptoms and putative biological mechanisms of HIV-CVD. It is plausible that immune activation and/or systemic inflammation may predict insomnia symptoms among PLWH, although human research examining this direction is limited [61]. Second, as previously mentioned, our insomnia variable was limited to a single item and did not comprehensively capture insomnia symptoms. Third, the examined biomarkers may not adequately capture the intricate and interconnected nature of the immune and coagulation processes involved in the development of HIV-CVD. However, the three biomarkers examined are among the most studied in relation to CVD risk [62], increasing our ability to make comparisons with past studies. Fourth, we were unable to adjust for some factors that may confound the relationship between insomnia symptoms and putative biological mechanisms of HIV-CVD (e.g., physical inactivity, obstructive sleep apnea). Future studies examining the mechanisms underlying the insomnia-CVD relationship among PLWH should ideally utilize a prospective design with a comprehensive assessment of insomnia (e.g., ISI, SCISD, DISP) and multiple indicators of immune activation, systemic inflammation, and coagulation.

In summary, we did not observe significant associations between insomnia symptoms and markers of immune activation, systemic inflammation, or coagulation. Our results raise the possibility that other mechanisms (e.g., hypertension or overweight/obesity) may be responsible for observed associations between insomnia symptoms and incident HIV-CVD. However, further research into the insomnia-biological mechanism relationships is still warranted as our study, and others in the extant literature, did not comprehensively assess insomnia. Ultimately, elucidating mechanistic pathways of the insomnia-CVD relationship in PLWH could determine the viability of insomnia as a potential behavioral treatment target for the management of chronic immune activation and the subsequent risk of CVD among PLWH.

## Acknowledgments

We thank the participants of the Veterans Aging Cohort Study.

## Author Contributions

**Conceptualization:** Brittanny M. Polanka, Samir K. Gupta, Tamika C. B. Zapolski, Adam T. Hirsh, Jesse C. Stewart.

**Data curation:** Suman Kundu.

**Formal analysis:** Suman Kundu.

**Funding acquisition:** Matthew S. Freiberg, Samir K. Gupta, Jesse C. Stewart.

**Project administration:** Matthew S. Freiberg.

**Supervision:** Jesse C. Stewart.

**Writing – original draft:** Brittanny M. Polanka.

**Writing – review & editing:** Brittanny M. Polanka, Suman Kundu, Kaku A. So-Armah, Matthew S. Freiberg, Samir K. Gupta, Tamika C. B. Zapolski, Adam T. Hirsh, Roger J. Bedimo, Matthew J. Budoff, Adeel A. Butt, Chung-Chou H. Chang, Stephen S. Gottlieb, Vincent C. Marconi, Julie A. Womack, Jesse C. Stewart.

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
