## [Decision Letter · Decision Letter 0]

30 Oct 2020

PONE-D-20-30184

Insomnia symptoms and biomarkers of monocyte activation, systemic inflammation, and coagulation in HIV: Veterans Aging Cohort Study

PLOS ONE

Dear Dr. Polanka,

Thank you for submitting your manuscript to PLOS ONE. After careful consideration, we feel that it has merit but does not fully meet PLOS ONE’s publication criteria as it currently stands. Therefore, we invite you to submit a revised version of the manuscript that addresses the points raised during the review process.

Both reviewers highly appreciated your manuscript and the quality of the presented data. I also enjoyed reading your paper for the novelty of the data and their potential impact to the field.

We look forward to receiving your revised manuscript.

Kind regards,

Cristian Apetrei, MD, PhD

Academic Editor

PLOS ONE

Journal Requirements:

2. Thank you for including your ethics statement:  "VACS-9 is a prospective, multisite, cohort study of HIV-positive veterans and age, sex, race/ethnicity, and clinical site-matched HIV-negative veterans from nine Department of Veterans Affairs (VA) medical centers across the U.S. [29, 30]. The VACS study was approved by the institutional review boards of each institution [29].".   

4. Thank you for stating the following in the Financial Disclosure section:

"Dr. Polanka is supported by the Ruth L. Kirschstein Institutional National Research Service Award (NIH T32HL007180-43) from the National Heart, Lung, and Blood Institute.Dr. Kundu reports no funding or conflict of interest. Dr. So-Armah reports funding from the National Institutes of Health and the Providence/Boston Center for AIDS Research. Dr. Freiberg reports funding from the National Institutes of Health. Dr. Zapolski reports funding from the National Institutes of Health (K01DA043654) and Indiana University. Dr. Hirsh reports funding from the National Institutes of Health and the U.S. Department of Veterans Affairs. Dr. Bedimo reports funding from the VA ORD, the National Institutes of Health, and ViiV Healthcare.  He also served on scientific advisory board for Merck & Co, and ViiV Healthcare. Dr. Gupta reports funding from the National Institutes of Health, Indiana University, and Gilead Sciences; advisory board fees from Gilead Sciences and GlaxoSmithKline/ViiV; and travel support to present data at scientific conferences from Gilead Sciences and Bristol-Myers Squibb. Dr. Budoff reports funding from the National Institutes of Health and General Electric. Dr. Butt reports funding from Gilead and Merck. Dr. Marconi received funding from the Emory University Center for AIDS Research (AI050409). Dr. Womack reports no funding or conflict of interest. Dr. Stewart reports funding from the National Institutes of Health and Indiana University. The Veterans Aging Cohort Study was funded by grant U10AA13566 from the National Institute on Alcohol Abuse and Alcoholism and Veterans Health Administration Public Health Strategic Health Core Group. This analysis was funded in part by grant R01HL126557 from the National Institutes of Health. The content is solely the responsibility of the authors and does not necessarily represent the official views of the Department of Veterans Affairs or the National Institutes of Health."

We note that you received funding from a commercial source: ViiV Healthcare, Gilead Sciences, GlaxoSmithKline, Bristol-Myers Squibb, General Electric, Merck.

Please provide an amended Competing Interests Statement that explicitly states these commercial funders, along with any other relevant declarations relating to employment, consultancy, patents, products in development, marketed products, etc.

Reviewers' comments:

Reviewer's Responses to Questions

**Comments to the Author**

1. Is the manuscript technically sound, and do the data support the conclusions?

Reviewer #1: Yes

Reviewer #2: Yes

2. Has the statistical analysis been performed appropriately and rigorously? 

Reviewer #1: Yes

Reviewer #2: Yes

3. Have the authors made all data underlying the findings in their manuscript fully available?

Reviewer #1: Yes

Reviewer #2: Yes

4. Is the manuscript presented in an intelligible fashion and written in standard English?

Reviewer #1: Yes

Reviewer #2: Yes

5. Review Comments to the Author

Reviewer #1: In this study, performed by Brittanny Polanka and co - workers among 1,542 people with HIV from the Veterans Aging Cohort Study Biomarker Cohort the authors carried out a thorough analysis with multiple variables, which allowed them to reach significant results concerning the lack of associations between insomnia symptoms and higher circulating levels of markers of monocyte activation, systemic inflammation, and coagulation in these patients.

Reviewer #2: PLWH have many co-morbidities that have yet to be fully mechanized and this is an important field to which researchers can contribute. The authors of the manuscript were interested in understanding whether insomnia is a risk factor in HIV cardiovascular disease and modeled insomnia data from the VACS Survey with various compounding risk factors to allow for proper adjustments. Unfortunately, their data did not support their hypothesis. Nonetheless, the authors presented their negative data with gusto and this reviewer applauds their discussion section, where the authors dove into potential reasons for why their hypothesis was not supported and other published data with mixed results.

Please see the following minor changes recommended:

Please revise people with human immunodeficiency virus (PWH) to people living with human immunodeficiency virus (PLWH) throughout the manuscript.

Correct line markers 113-118 overlapping with Table 1.

Line 251: “our samples high prevalence” should be “our sample’s high prevalence”

6. PLOS authors have the option to publish the peer review history of their article (what does this mean?). If published, this will include your full peer review and any attached files.

Reviewer #1: No

Reviewer #2: No

---

## [Author Response · Author response to Decision Letter 0]

22 Dec 2020

Please see the point-by-point response letter labeled “Response to Reviewers” included in the online submission describing how we addressed each of the reviewer/journal comments.

---

## [Editor Report · Decision Letter 1]

13 Jan 2021

Insomnia symptoms and biomarkers of monocyte activation, systemic inflammation, and coagulation in HIV: Veterans Aging Cohort Study

PONE-D-20-30184R1

Dear Dr. Polanka,

We’re pleased to inform you that your manuscript has been judged scientifically suitable for publication and will be formally accepted for publication once it meets all outstanding technical requirements.

Kind regards,

Cristian Apetrei, MD, PhD

Academic Editor

PLOS ONE
---

## [Editor Report · Acceptance letter]

29 Jan 2021

PONE-D-20-30184R1 

Insomnia symptoms and biomarkers of monocyte activation, systemic inflammation, and coagulation in HIV: Veterans Aging Cohort Study 

Dear Dr. Polanka:

I'm pleased to inform you that your manuscript has been deemed suitable for publication in PLOS ONE. Congratulations! Your manuscript is now with our production department. 

Kind regards, 

on behalf of

Dr. Cristian Apetrei 

Academic Editor

PLOS ONE